

# Detecting high-emitting methane sources in oil/gas fields using satellite observations

Daniel H. Cusworth[1,2], Daniel J. Jacob[1,2], Jian-Xiong Sheng[2], Joshua Benmergui[2], Alexander J. Turner[3], Jeremy Brandman[4], Laurent White[4], and Cynthia A. Randles[4]

[1]Department of Earth and Planetary Sciences, Harvard University, Cambridge, 02138, USA
[2]School of Engineering and Applied Sciences, Harvard University, Cambridge, 02138, USA
[3]College of Chemistry/Department of Earth and Planetary Sciences, University of California, Berkeley, CA, USA
[4]ExxonMobil Research and Engineering Company, Annandale, NJ, USA

*Correspondence to*: Daniel Cusworth (dcusworth@fas.harvard.edu)

**Abstract.** Methane emissions from oil/gas fields originate from a large number of relatively small and densely clustered point sources. A small fraction of high-mode emitters can make a large contribution to the total methane emission. Here we conduct observation system simulation experiments (OSSEs) to examine the potential of recently launched or planned satellites to detect and locate these high-mode emitters through measurements of atmospheric methane columns. We

simulate atmospheric methane over a generic oil/gas field (20-500 production sites of different size categories in a $50 \times 50$ km$^2$ domain) for a 1-week period using the WRF-STILT meteorological model with $1.3 \times 1.3$ km$^2$ horizontal resolution. The simulations consider many random realizations for the occurrence and distribution of high-mode emitters in the field by sampling bi-modal probability density functions (pdfs) of emissions from individual sites. The atmospheric methane fields for each realization are observed virtually with different satellite and surface observing configurations. Column methane

enhancements observed from satellites are small relative to instrument precision, even for high-mode emitters, so an inverse analysis is necessary. The inverse analysis can be regularized effectively using a L-1 norm to provide sparse solutions for a bi-modally distributed variable. We find that the recently launched TROPOMI instrument (low Earth orbit, $7 \times 7$ km$^2$ nadir pixels, daily return time) and the planned GeoCARB instrument (geostationary orbit, $2.7 \times 3.0$ km$^2$ pixels, $2\times$ or $4\times$/day return time) are successful at locating high-emitting sources for fields of 20-50 emitters within the $50 \times 50$ km$^2$ domain but

are unsuccessful for denser fields. GeoCARB does not benefit significantly from more frequent observations ($4\times$/day vs. $2\times$/day) because of temporal error correlation in the inversion. It becomes marginally successful when allowing a 5-km error tolerance for localization. A next-generation geostationary satellite instrument with $1.3 \times 1.3$ km$^2$ pixels, hourly return time, and 1 ppb precision can successfully detect and locate the high-mode emitters for a dense field with up to 500 sites in the $50 \times 50$ km$^2$ domain. The capabilities of TROPOMI and GeoCARB can be usefully augmented with a surface air observation

network of 5-20 sites, and in turn the satellite instruments increase the detection capability that can be achieved from the surface sites alone.



# 1 Introduction

Anthropogenic methane emissions from oil/gas fields originate from a large number of relatively small and densely clustered point sources (Allen et al., 2013). For example, the Barnett Shale in Texas has over 20000 well pads spread over a $300 \times 300$ km$^2$ domain, contributing 40% of total oil/gas emissions from the region (Lyon et al., 2015). Seven percent of the wells

contribute 50% of total well emissions (Rella et al., 2015; Zavala-Araiza et al., 2015). Identifying such high-emitting wells is of both economic and environmental interest. We present here observing system simulation experiments (OSSEs) to examine the potential of using satellite observations of atmospheric methane for this purpose.

Satellites measure atmospheric columns of methane by backscattered solar radiation in the shortwave infrared (SWIR), with near uniform sensitivity down to the surface under clear-sky conditions (Jacob et al., 2016). The satellite

record for SWIR methane began with the SCIAMACHY instrument (2003-2012; Frankenberg et al., 2005), which provided coarse resolution measurements (30 x 60 km$^2$ in nadir). The currently operating GOSAT instrument (2009-; Kuze et al., 2016) has finer resolution (10-km diameter pixels) but sparse coverage (250 km apart). The TROPOMI instrument launched in October 2017 provides daily measurements at $7 \times 7$ km$^2$ nadir resolution (Hu et al., 2018). The geostationary GeoCARB instrument, to be launched in the early 2020s, is currently planned to provide $2.7 \times 3$ km$^2$ pixel resolution with a return time

that may range from one to four times per day (Polonsky et al., 2014; O'Brien et al., 2016). Other geostationary methane satellite missions have been proposed with various combinations of more frequent coverage, finer pixel resolution, and higher instrument precision (Fishman et al., 2012; Butz et al., 2015; Xi et al., 2015; Propp et al., 2017).

A number of studies have examined the value of satellite observations for quantifying methane sources. Inverse analyses of SCIAMACHY and GOSAT data have focused on quantifying emissions at ~100 km regional scales

(Bergamaschi et al., 2013; Wecht et al., 2014a; Alexe et al., 2015; Turner et al., 2015). OSSEs have shown the potential for TROPOMI and GeoCARB to effectively constrain emissions at the 25-100 km scale without the multiyear averaging required by SCIAMACHY and GOSAT (Wecht et al., 2014b; Sheng et al., 2018a). Other OSSEs have examined the potential for satellites to quantify large point sources from plume observations (Buchwitz et al., 2013; Rayner et al., 2014; Varon et al., 2018). A recent study by Turner et al. (2018) evaluated the capability of TROPOMI and GeoCARB to quantify

emissions in the Barnett Shale down to the kilometer scale for a 1-week observing period. They found that GeoCARB should have some capability for constant sources over a 1-week period but not for transient sources. Hase et al. (2017) simulated surface and aircraft pseudo-observations over North America and used them to constrain North American emissions at 1° × 1° resolution. They found that sparse optimization better constrained local methane hotspots than the standard Bayesian approach.

Here we target a different problem. Given a dense population of production sites (wells) in an oil/gas field, can satellites localize high-mode emitters to enable corrective action? In this problem, quantifying emissions is not as important as identification of the high-mode emitters. The location of the individual point sources is known, but their mode of emission (normal low-mode or high-mode) is unknown. Once a well starts emitting in the high mode, it continues doing so until



corrective action is taken. Satellites offer an attractive monitoring approach for identifying high-mode emitters but their capability may be limited by return frequency, cloud cover, pixel resolution, error in the atmospheric transport model needed to relate the plume to the location of emission, and limitations in the inverse method for identifying sparse high-mode sources. Here we will evaluate the potential of different satellite observing configurations and inverse methods to address

this problem with application to TROPOMI, GeoCARB, and finer-resolution geostationary data. We will also examine whether the information from satellites can be usefully complemented with a supporting network of surface observations.

## 2 Observing System Simulation Experiment

We consider a hypothetical oil/gas field of dimension $50 \times 50$ km$^2$ with 20, 50, 100, or 500 randomly placed production sites

(wells), corresponding to site densities of 0.008 km$^{-2}$, 0.02 km$^{-2}$, 0.04 km$^{-2}$, and 0.2 km$^{-2}$, respectively. The latter case corresponds to the average site density in the Barnett Shale. We create a large ensemble of emission scenarios in each case where different random subsets of sites of different production size categories (small: 10-100 million cubic feet per day (Mcf/d); medium: 100-1000 Mcf/d; large: 1000+ Mcf/d) are in the high-emission mode, and we simulate the resulting atmospheric methane concentration fields with the WRF meteorological model at $1.3 \times 1.3$ km$^2$ resolution. We then sample

this pseudo-atmosphere with different satellite and surface observing configurations and apply different inverse methods to detect the high emitters. Detection success is evaluated for each observing configuration and inverse method using statistics for the ensemble of emission scenarios. We describe in this section the different elements of the OSSE.

### 2.1 Constructing an ensemble of emission fields

Production sites within the $50 \times 50$ km$^2$ domain are randomly placed on the $1.3 \times 1.3$ km$^2$ WRF model grid, with at most one

site per grid cell. Emission statistics for the sites are based on observations from the Barnett Shale Coordinated Campaign (Lyon et al. 2015). For each scenario we randomly assign a production size category to each site with 23% of the sites as small, 62% as medium, and 15% as large (Rella et al., 2015). We then assign an emission rate for each site by randomly sampling the bi-modal probability density functions (pdfs) describing low-mode emissions and high-mode emissions for each size category (Lan et al., 2015; Rella et al., 2015; Yacovitch et al., 2015). We assume no other sources in the domain.

Figure 1 shows the pdfs of methane emissions for each production site size category. We flag production sites to be in the high-emission mode if they exceed an emission threshold of 40 kg h$^{-1}$ (axis break in Figure 1), which corresponds on average to 5% of all the sites. High-mode emissions from small facilities are much lower, centered around 24 kg h$^{-1}$, and would be difficult to distinguish from the normal (low) emission mode. Thus we do not attempt to detect them as high-mode emitters.

Figure 2 shows a sample realization of the oil/gas field with 24 small production sites, 67 medium sites, and 9 large sites (100 total) within the $50 \times 50$ km$^2$ domain. In this realization there are five sites in the high-emission mode. We



generate 500 emission scenarios in the same fashion as Figure 2 by randomly assigning size categories for each site (small, medium large) and randomly sampling the emission pdfs from Figure 1.

## 2.2 Constructing pseudo-observations of atmospheric methane

We use the meteorological simulation previously generated by Turner et al. (2018) for a 1-week period (October 19-25, 2013) in the Barnett Shale. This simulation applied the Weather Research and Forecasting Model (WRF; Skamarock et al., 2008) at 1.3 km horizontal resolution to drive the Stochastic Time-Inverted Lagrangian Transport (STILT) model (Nehrkorn et al., 2010). STILT is a receptor-oriented Lagrangian particle dispersion model that defines the source footprints for individual atmospheric observations. Turner et al. (2018) applied it to generate $1.3 \times 1.3$ km$^2$ hourly footprints for any daytime surface or atmospheric column observation in a $70 \times 70$ km$^2$ domain. The column footprints were weighted with a typical near-uniform SWIR averaging kernel for satellite observations (Worden et al., 2015). We use their ensemble of footprints and add to it hourly footprints for surface observations at night. The $70 \times 70$ km$^2$ observing domain encompasses our $50 \times 50$ km$^2$ oil/gas field plus 10 km outside the boundaries (Figure 2) to account for plume transport.

The $70 \times 70$ km$^2$ archive of WRF-STILT footprints allows us to immediately compute the time-dependent methane concentration field associated with any emission scenario. Figure 3 shows a sample footprint, expressing the sensitivity of atmospheric concentrations at a given location and time $i$ to the emission field upwind. Column footprints are about an order of magnitude smaller than surface footprints because of the dilution effect of the column measurement. Taking the footprints to represent the true atmospheric transport relating emissions to atmospheric concentrations for that location and time, we can combine them with any realization of our emission field (Section 2.1) to generate the true time-dependent methane concentrations in the domain to be sampled by the instruments.

Satellite observations of methane column concentrations are conventionally expressed in unit of dry column mean mixing ratio (ppb), which is the ratio of the vertical column density of methane to the vertical column density of dry air (Jacob et al., 2016). The footprint for location and time $i$ is mathematically represented as $\mathbf{h}_i = (\partial y_i / \partial \mathbf{x})^T$ (units ppb $\mu$mol$^{-1}$ m$^2$ s$^1$) where $y_i$ is the methane concentration (ppb) for that location and time, and $\mathbf{x}$ ($\mu$mol m$^{-2}$ s$^{-1}$) is a vector of dimension $n$ describing the gridded emission field for the $n$ emitters in the domain. From the archived footprints $\mathbf{h}_i$ covering the complete set of observing locations and times, the true atmospheric concentration can be immediately constructed for any emission field $\mathbf{x}$ as $y_i = \mathbf{h}_i \bullet \mathbf{x} + b$, where $\bullet$ denotes the scalar product and $b$ is a background assumed here to be constant.

A given methane observing configuration makes $m$ observations of the domain over the 1-week simulation period. The true methane concentrations for that observation ensemble can be assembled as an $m$-dimensional vector $\mathbf{y_{true}} = \mathbf{Hx}$ where $\mathbf{H} = \partial \mathbf{y_{true}} / \partial \mathbf{x}$ is the $m \times n$ Jacobian matrix of footprints with rows $\mathbf{h}_i^T$. The pseudo-observations are then generated as $\mathbf{y} = \mathbf{y}_{true} + \sigma \boldsymbol{\varepsilon}$ where $\sigma$ is the instrument precision (one standard deviation) and the vector $\boldsymbol{\varepsilon}$ is a random realization of Gaussian noise with mean value of zero and standard deviation of unity for each vector element.




## 2.3 Satellite and surface observing configurations

Table 1 describes the different satellite observing configurations evaluated in this work including TROPOMI, GeoCARB with 2 or 4 return times per day, and an aspirational next-generation geostationary instrument with $1.3 \times 1.3$ km$^2$ pixel resolution, 1 ppb precision, and hourly return frequency between 8 and 17 local time (LT). Clouds would interfere with

satellite observations but here we assume clear-sky conditions to simplify the discussion.

We also wish to determine the benefit of a well-positioned surface air monitoring network for supplementing the satellite observations. Assume that we have $M$ fixed monitoring instruments to deploy measuring surface air methane concentrations *in situ*. We want to place them in a configuration that maximizes the information that they would provide, assuming an isotropic wind for generality. A trivial solution would be to place an instrument at each production site, in

which case the monitoring problem would be fully solved, but this solution may not be practical for a large number of production sites. Given a known spatial distribution of emitters (the locations of the production sites), we use the $k$-means spatial clustering approach (Hartigan and Wong, 1979) to select monitoring site locations minimizing the distances to emitter locations. Figure 2 shows the selected locations for five surface monitoring sites. We assume that these sites report hourly data with 1 ppb precision.

An important consideration in the interpretation of satellite observations is that methane column enhancements from point sources are typically small relative to instrument precision, even in the high-emitting mode. This reflects the general problem of methane emissions originating from a large number of relatively small point sources (Jacob et al., 2016; Varon et al., 2018). Figure 4 shows the pixel-resolved distribution of atmospheric methane column enhancements above the background for a single pass of the different satellite instruments sampling the emission field of Figure 2. The enhancements

are less than 1 ppb even for 1.3 km pixels and are weaker at coarser pixel resolution. This is less than the single-scene precision of the satellite instruments (Table 1). Successful detection of high-mode emitters thus requires the sampling of many pixels, across the plume and/or through repeated sampling, to reduce the noise. This is less of an issue for surface air measurements, where methane enhancements are an order of magnitude higher (Figure 3). On the other hand, surface monitoring sites are spatially sparse. For both satellite and surface air observations, a formal inverse analysis of the ensemble

of atmospheric observations accounting for plume transport is required for detection of the high-mode emitters.

## 2.4 Inverse methods

Given a set of observations $\mathbf{y}$ and Jacobian matrix $\mathbf{H}$, we need an inverse method to determine the best solution ($\hat{\mathbf{x}}$) of the emission field $\mathbf{x}$ at predetermined locations. The inversion should be able to detect the small fraction of sources in the high-

emitting mode, with detection being more important than quantification. This is known as a sparse-solution problem, where most elements of the emission field $\mathbf{x}$ are very small (for which an optimized value of zero would be acceptable), and a few of the elements are relatively large. We use regularized least squares regression (Evgeniou et al., 2000), also known as Tikhonov Regularization, where the solution is found by minimizing the cost function $J(\mathbf{x})$,



$$J(\mathbf{x}) = (\mathbf{Hx} - \mathbf{y})^{\mathrm{T}} \mathbf{R}^{-1}(\mathbf{Hx} - \mathbf{y}) + \lambda \|\mathbf{x}\|_L \quad (1)$$

Here the first term on the right hand side represents the ordinary least-squares cost function, such that the solution would

minimize the residuals between the prediction $\mathbf{Hx}$ and the observations weighted by the observational error covariance matrix $\mathbf{R}$. The second term represents a norm of $\mathbf{x}$, or a measure of the magnitude of the vector $\mathbf{x}$, with $\lambda$ an adjustable parameter. Adding this second term in the cost function penalizes the total magnitude of $\mathbf{x}$ in the solution, which reduces overfitting to noise and regularizes the solution. When $L = 1$, this is known as L-1 regularization or the least absolute shrinkage and selection operator (LASSO; Tibshirani, 1996), and Equation 1 takes the form:

$$J(\mathbf{x}) = (\mathbf{Hx} - \mathbf{y})^{\mathrm{T}} \mathbf{R}^{-1}(\mathbf{Hx} - \mathbf{y}) + \lambda \sum_{k=1}^{n} |x_k| \quad (2)$$

When $L = 2$, Equation 1 takes the form known as L-2 regularization or Ridge Regression (Evgeniou et al., 2000):

$$J(\mathbf{x}) = (\mathbf{Hx} - \mathbf{y})^{\mathrm{T}} \mathbf{R}^{-1}(\mathbf{Hx} - \mathbf{y}) + \lambda \mathbf{x}^T \mathbf{x} \quad (3)$$

Equation 3 is equivalent to the standard Bayesian optimization (Rodgers, 2000) assuming Gaussian distributions, a prior emission estimate of zero, and uniform prior error variance of $\lambda^{-1}$.

The observational error covariance matrix $\mathbf{R} = (r_{ij})$ accounts for both instrument and model transport errors. The diagonal terms add the corresponding error variances in quadrature:

$$r_{ii} = \sigma_I^2 + \sigma_M^2 \quad (4a)$$

where $\sigma_I$ is the instrument error standard deviation as given by the precision in Table 1, and $\sigma_M$ is the model transport error standard deviation previously estimated to be 4 ppb (Turner et al., 2018). We use the same model transport error for surface concentrations and satellite columns. Off-diagonal terms account for model transport error correlation between different observations. Following Turner et al. (2018), we assume a temporal error correlation length scale ($\tau$) of 2 hours and a spatial error correlation length scale ($\ell$) of 40 km:

$$r_{ij} = \sigma_M^2 \times \exp\left\{-\frac{d}{\ell}\right\} \exp\left\{-\frac{t}{\tau}\right\} \quad \text{for } i \neq j \quad (4b)$$

where $d$ and $t$ are the distance and elapsed time between observations $y_i$ and $y_j$.

Additional model transport error correlation applies when combining satellite and surface air observations in the inversion, since the footprints can be similar (Figure 3). To quantify this error correlation, we use the work of Sheng et al.





(2018b), who jointly compared column (TCCON) and surface air (NOAA) measurements of methane at Lamont, Oklahoma with GEOS-Chem transport model simulations. By correlating the coincident model-observation differences for the column ($i$) and surface air ($j$) observations we find a model transport error correlation coefficient cor($i, j$) = 0.65 that we apply to the corresponding off-diagonal terms:

$$r_{ij} = \text{cor}(i, j) \times \sigma_M^2 \times \exp\left\{-\frac{d}{\ell}\right\} \exp\left\{-\frac{t}{\tau}\right\} \quad (4c)$$

Inverse solutions derived using L-1 regularization produce sparser solutions than the L-2 counterpart (Tibshirani, 1996), which is desirable for our application and has previously been shown to produce good results for constraining methane hotspots (Hase et al. 2017). Here we will perform both L-1 and L-2 inversions and compare the results.
Minimization of $J(\mathbf{x})$ in Equations 2 and 3 to obtain the solution $\hat{\mathbf{x}}$ corresponding to $dJ/d\mathbf{x} = 0$ is done numerically using coordinate gradient descent (Friedman et al., 2009). The regularization parameter $\lambda$ is chosen so that the mismatch between model and observations is small, but not so small that the solution $\hat{\mathbf{x}}$ is overfit to random noise, which would occur when $\lambda =$ 0. We use the process of 5-fold cross-validation to select an optimal $\lambda$ value (Arlot and Celisse, 2010). This process randomly samples **H** and **y** into a training and validation set. Minimization of $J$ is done on the training set using an array of $\lambda$
values. The process is repeated five times, and the value of $\lambda$ that on average minimizes the residual error on the validation set is retained.

**2.5 Detection of high-emission modes**

Success in the detection of high-mode emitters from the distribution of $\hat{\mathbf{x}}$ can be determined by comparison to the actual
occurrence and location of these emitters as defined in Section 2.1 and illustrated in Figure 2. In a real-world application we would not know the actual pdfs of emissions (Figure 1), so we need to diagnose the occurrence of high-mode emitters on the basis of anomalies in the distribution of $\hat{\mathbf{x}}$. We define high-mode elements as being more than $S$ standard deviations from the mean of the $\hat{\mathbf{x}}$ distribution, where $S$ is varied in the 1.65-2.5 range to examine the associated sensitivity.

The detection of high-mode emitters by the inversion is graded into four categories: 1) true positives (TP), or the
inversion correctly identifying the locations of the high-mode emitters, 2) true negatives (TN), or the inversion correctly identifying the locations of the low-mode emitters, 3) false positives (FP), or the inversion signaling a high-mode emitter when in reality the emitter is in the low mode, and 4) false negatives (FN), or the inversion signaling a low-mode emitter when in reality the emitter is in the high mode.

We compile these grades into three overall performance metrics (Brasseur and Jacob, 2017). The probability of
detection (POD) is defined as the ratio of true positives to true positives plus false negatives:

$$POD = \frac{\Sigma\, TP}{\Sigma\, TP + \Sigma\, FN}$$



This metric measures the ability to detect high-mode emitters. The false alarm ratio (FAR) is defined as the ratio of false positives to false positives plus true positives:

$$FAR \ = \ \frac{\Sigma\, FP}{\Sigma\, TP + \Sigma\, FP}$$

This metric measures the reliability of high-mode emission occurrences detected by the inversion.

A perfect observing system would have a POD of one and a FAR of zero. Here we define a successful observing system as achieving a POD of 0.8 (80%) and a FAR of 0.2 (20%). These criteria, although somewhat arbitrary, allow us to succinctly summarize the success of each observing configuration.

We combine the POD and FAR metrics into one overall performance metric called the Equitable Threat Score (ETS; Wang,
10   2014):

$$ETS = \ \frac{\Sigma\, TP - \alpha}{\Sigma\, TP + \Sigma\, FP + \Sigma\, FN - \alpha}$$

where $\alpha$ is the number of TP predictions that are expected by chance:

$$\alpha = \frac{(\Sigma\, TP + \Sigma\, FP)(\Sigma\, TP + \Sigma\, FN)}{\Sigma\, TP + \Sigma\, FP + \Sigma\, FN + \Sigma\, TN} = \frac{1}{N} \times \frac{\Sigma\, FP}{FAR} \times \frac{\Sigma\, TP}{POD}$$

and $N = \Sigma\, TP + \Sigma\, FP + \Sigma\, FN + \Sigma\, TN$. The ETS measures how well the high-mode emitters detected by the observing system correspond to the actual occurrences, beyond what could be achieved by chance. A perfect observing system has an ETS of one, and a system performing worse than chance would have a negative ETS. An observing system with POD of 0.8 and FAR of 0.2 has an ETS of 0.65 for a field where 5% of emitters are in the high mode. We take this as our ETS criterion for successful detection.

## 3 Results and discussion

### 3.1 Performance of different satellite and surface observing systems

We begin by testing the ability of each satellite configuration of Table 1 to detect high-mode emitters from fields of 20 to 500 randomly scattered production sites within the 50×50 km² domain. For a given number of sites, we conduct each test for
500 different realizations of the emission field assigning randomly each production site to a size category (small, medium, large) and sampling randomly the pdfs of Figure 1. Emitter locations are fixed across all 500 realizations. Figure 5 shows the POD, FAR, and ETS results for a field of 100 emitters and compares the results of L-1 and L-2 regularizations. The values represent the mean results for the ensemble of 500 realizations, and the error bars represent the range of results when the high-mode detection threshold $S$ is varied from 1.65 to 2.5. We find that L-1 regularization provides better predictions for all
cases. This is especially the case for the next-generation satellite, where L-1 regularization produces a POD of 0.85 with a near-perfect FAR of 0.04. L-2 regularization is more conducive to spreading emissions across a broader array of state vector



elements. The better performance of L-1 regularization is also observed for other site densities (not shown). We use L-1 regularization in what follows.

Figure 5 also compares the performances of the satellite observing systems to those of an ensemble of 5-20 optimally placed (*k*-means) surface sites. We find that the surface observing system performs comparably to GeoCARB. We explore combining satellite and surface observations into a single prediction in Section 3.3.

The results from Figure 5 show that TROPOMI and GeoCARB are unsuccessful in locating high-mode emitters for a 100 site field. We examine the sensitivity of this result to site density. Figure 6 compares the detection results for fields of 20, 50, 100, and 500 production sites within the $50 \times 50$ km$^2$ domain. For a field of only 20 emitters, TROPOMI is successful and GeoCARB produces near perfect results. For a field of 50 emitters, TROPOMI is no longer successful, but GeoCARB is still marginally successful due to finer pixel resolution and higher instrument precision. We find in general that GeoCARB gains little by sampling four times a day ($4 \times$/day) vs. $2 \times$/day. This is due to the temporal model error correlation between successive GeoCARB observations. Accounting for cloud cover would show more benefit from $4 \times$/day observations, since a higher frequency of observations allows a greater chance of sampling clear-sky conditions, although the benefit depends on the cloud persistence time scale (Sheng et al., 2018a).The ability of a satellite observing configuration to localize high-mode emitters thus depends not only on repeat time, resolution, and precision, but also on the density of emitters within a field. For the high-density fields of 100 and 500 emitters we find that only the next-generation satellite instrument is successful. Detecting individual high-mode emitters in much denser fields would require geostationary satellite observations with sub-km pixels but this is beyond the scope of current proposals.

**3.2 Spatial tolerance in detection of high-mode emitters**

The results from Figure 6 are somewhat pessimistic regarding the ability of near-future satellite observations (TROPOMI and GeoCARB) to detect the locations of high-mode emitters in dense emission fields. It may be acceptable to relax the localization criterion. If the observing system detects a false positive that is sufficiently close to the actual location of a high-mode emitter, then the detection may still have some value. In our OSSE setup, localization is effectively limited by the 1.3 km grid resolution of the WRF simulation. To examine the sensitivity to localization, we repeated the analysis allowing for 3-5 km tolerance of false predictions. Figure 7 shows the results for a field of 100 emitters. We find that spatial tolerance significantly improves the performance of GeoCARB but still falls short of our success criterion. The FAR decreases below 0.2 for 3 km tolerance and below 0.1 for 5 km tolerance, but the POD only improves to 0.7 and thus the ETS remains below 0.65.

**3.3 Combining satellite and surface observations**

We saw in Section 3.1 that only the next-generation satellite instrument can successfully detect high-mode emitters when the site density is high. Here we examine if a combination of satellite and surface observations can improve detection, i.e., if



TROPOMI and GeoCARB could benefit from an *in situ* supporting surface network and vice versa. This is addressed with a joint inversion of the satellite and surface observations, taking into account the error correlation between the two as described in Section 2.4.

Figure 8 shows the results for a field of 100 emitters. The already successful next-generation instrument shows no
benefit from added surface sites. On the other hand, the surface sites provide greatly added value to TROPOMI and GeoCARB. Adding 10-20 surface sites enables near-successful detection of the high-mode emitters. At the same time, TROPOMI and GeoCARB data add significantly to the performance of a surface observing system alone by providing observations with more spatial coverage. We find that TROPOMI and GeoCARB perform similarly when added to surface sites, and that their main benefit is to decrease the FAR. Accounting for clouds would show more benefit for GeoCARB
because the finer pixels allow for more frequent clear-sky observations (Sheng et al., 2018a).

## 4 Conclusions

We performed observing system simulation experiments (OSSEs) to test the ability of near-future satellite instruments measuring atmospheric methane (TROPOMI, GeoCARB, next-generation geostationary) to detect high-mode point source
emitters among a dense field of individual point sources, alone or supported by a surface monitoring network. We focused on the practical problem of detecting high-mode emitters in an oil/gas production field with a high density of wells. Remote detection from satellites, combined with operator knowledge, could supplement on-site leak detection and repair (LDAR) programs to identify and fix unexpected high emitters. Our results can be summarized usefully in terms of answers to questions that a field manager might have:

*Can I rely on satellite data alone to detect high-mode emitters among the production sites in my oil/gas field?*
We find that TROPOMI and GeoCARB can detect high-mode emitters as long as the density of point sources is relatively small (20 sites within our $50 \times 50$ km$^2$ domain, or a density of 0.008 km$^{-2}$). GeoCARB shows little difference in success rate for 2 or 4 overpasses per day. GeoCARB is marginally successful for 50 sites (0.02 km$^{-2}$) but fails for 100 sites (0.04 km$^{-2}$).
A next-generation geostationary satellite instrument with ~1-km pixel resolution and hourly return time would deliver precise detection in dense fields up to 500 sites (0.2 km$^{-2}$). Allowing for a 5-km spatial error tolerance for localization, we find that GeoCARB comes close to successful detection in a field of 100 sites.

*How should I analyze the satellite observations to detect high-mode emitters?*
Detection of high-mode emitters from satellite observations is not a simple matter of flagging hot spots because the methane column enhancements are typically small compared to instrument precision, even for high-mode emitters. Repeated observation combined with inverse analysis using an atmospheric transport model is needed. We find that an inversion with




L-1 regularization produces better results than L-2 regularization. This is expected since the L-1 regularization method is designed to recover sparse signals.

*Can I usefully supplement satellite information with surface monitoring?*

Both TROPOMI and GeoCARB add significantly to the information provided by a surface monitoring network of 5-20 sites within the $50 \times 50$ km$^2$ domain, and both benefit from the added surface information. The combination of these satellite instruments with the surface monitoring instruments can deliver successful detection of high-mode emitters through a joint inversion. Adding surface sites provides no benefit to the next-generation geostationary instrument, which can successfully detect high-mode emitters on its own as long as skies are clear.

**Acknowledgments**. This work was supported by the ExxonMobil Research and Engineering Company, the US Department of Energy (DOE) Advanced Research Projects Agency – Energy (ARPA-E), and the NASA Earth Science Division. This research used the Savio computational cluster resource provided by the Berkeley Research Computing program at the University of California, Berkeley (supported by the UC Berkeley Chancellor, Vice Chancellor for Research, and Chief Information Officer). This research also used resources from the National Energy Research Scientific Computing Center, which is supported by the Office of Science of the U.S. Department of Energy under Contract No. DE-AC02-05CH11231.

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



**Table 1**. Observing configurations considered in this work.

| Instrument | Observation Frequency | Pixel size ($km^2$) | Precision[a] (ppb) | Number of observations[b] |
|---|---|---|---|---|
| *Satellites* | | | | |
| TROPOMI | Daily[c] | 7.0 x 7.0 | 11[d] | 567 |
| GeoCARB 2×/day | 2× daily[e] | 2.7 x 3.0 | 4.0 [f] | 7700 |
| GeoCARB 4×/day | 4× daily[g] | 2.7 x 3.0 | 4.0 | 15400 |
| Next generation[h] | Hourly[i] | 1.3 x 1.3 | 1.0 | 164500 |
| *Surface sites*[j] | Hourly[k] | point | 1.0 | 840 – 3360[l] |

[a]Dry column mean mixing ratio for the satellite observations
[b]One week of clear-sky conditions in the $70 \times 70$ $km^2$ domain
[c]13 local solar time (LT)
[d]Butz et al. (2012)
[e]12 and 16 LT
[f]O'Brien et al. (2016)
[g]10, 12, 14, and 16 LT
[h]Aspirational instrument combining the characteristics of instruments currently at the proposal stage (Fishman et al., 2012; Butz et al., 2015; Xi et al., 2015)
[i]Between 8 and 17 LT
[j]Observing *in situ* surface air concentrations
[k]Day and night
[l]For 5 to 20 surface monitoring sites





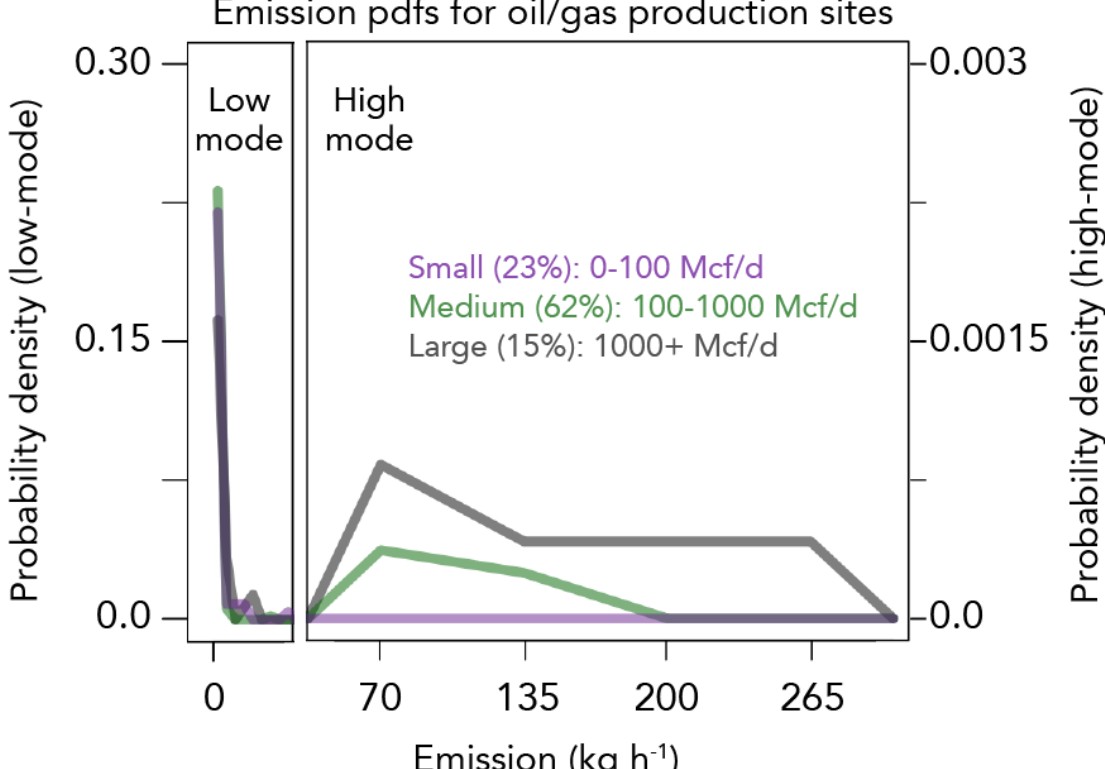

**Figure 1**. Probability density functions (pdfs) of emissions for oil/gas production sites of different production size categories (small, medium, and large), taken from Barnett Shale observations (Lan et al., 2015; Rella et al., 2015; Yacovitch et al., 2015). Note the difference in y-axis scales between the left (low-mode) and right (high-mode) panels. The axis break at 40 kg h$^{-1}$ represents the threshold for flagging an emitter as high.



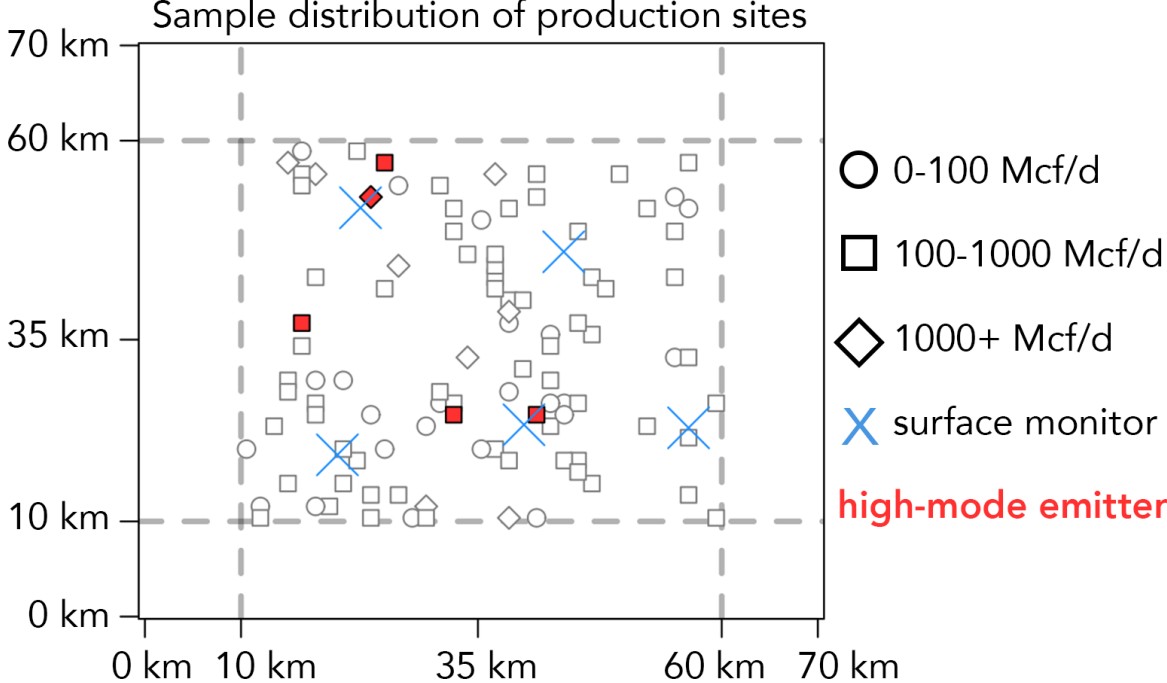

5 **Figure 2**. Sample realization of emissions from a hypothetical oil/gas production field with 100 production sites of different production size categories (symbols) within a 50×50 km² domain (dashed line). Different production size categories are shown with symbols. Red shading indicates high-mode emitters. Blue symbols mark the locations of five surface air monitoring sites placed according to the *k*-means algorithm.





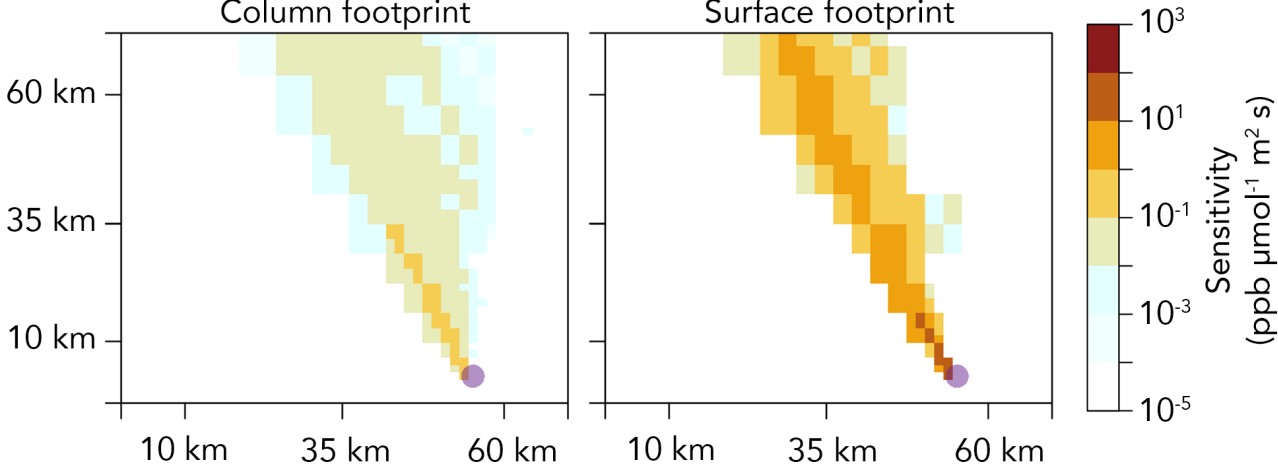

**Figure 3**: Sample sensitivities of observed atmospheric concentrations (column and surface) to surface emissions upwind, defining the emission footprint for that observation. Values are shown here for a particular observation point (purple dot) and time (October 19, 2013 at 09 LT). Concentrations are in mixing ratio units of ppb (dry column mean mixing ratio for the column) and emissions are in units of $\mu$mol m$^{-2}$ s$^{-1}$.





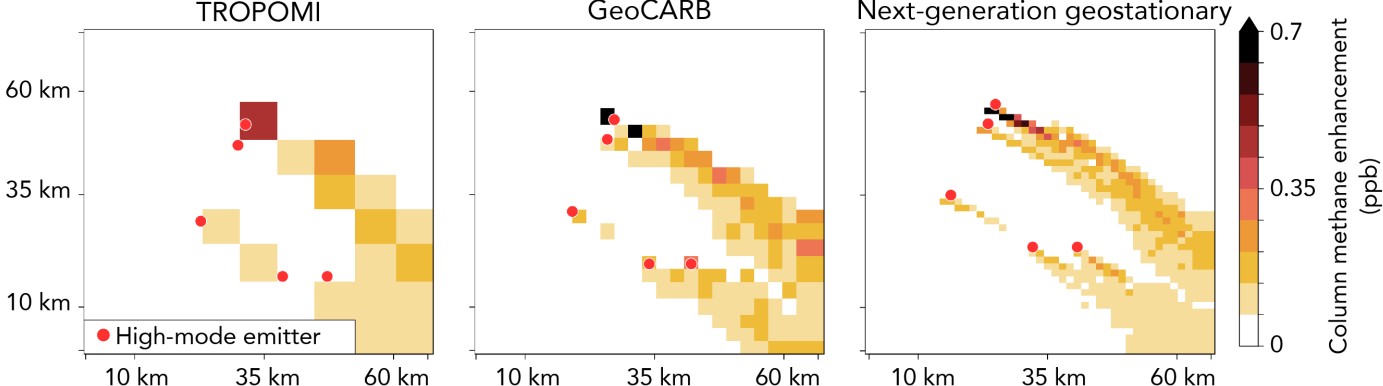

**Figure 4**: Simulated noiseless methane column enhancement for sampling by single overpasses of TROPOMI, GeoCARB, and a next-generation high-resolution geostationary satellite (Table 1). Emission field is that of Figure 2. The locations of the five high-mode emitters in that field are indicated. Values are for 22 October 2013 at 13 LT.



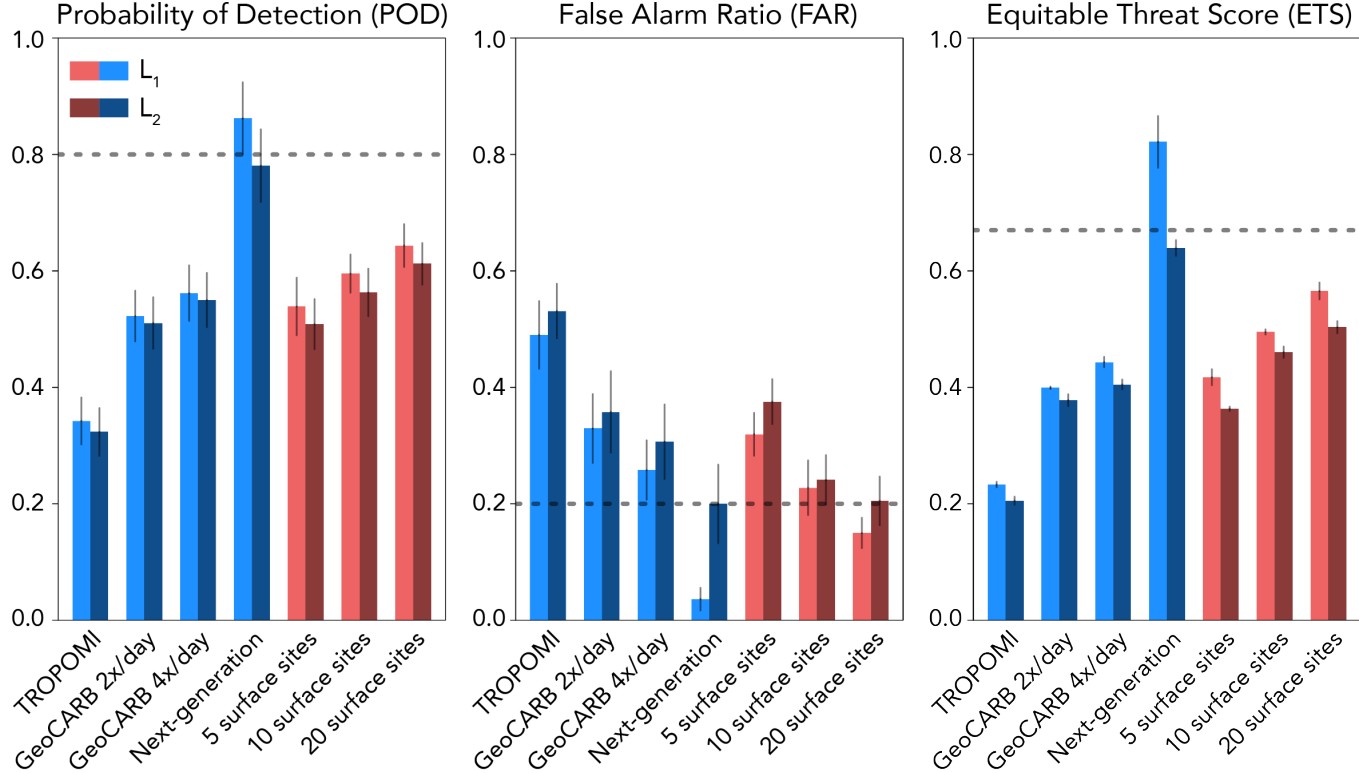

**Figure 5**. Probability of detection (POD), false alarm ratio (FAR), and Equitable Threat Score (ETS) of high-mode emitters for each satellite and surface observing configuration. Each bar represents the mean of 500 observing system simulation experiments (OSSEs), where 100 production sites in a $50 \times 50$ km$^2$ domain were used to construct 500 random realizations of an emission field including different subsets of high-mode emitters. For each observing configuration, the left bar (lighter color) shows results for the inversion with L-1 regularization, and the right bar (darker color) is for the L-2 regularization. The dashed lines represent the POD, FAR, and ETS criteria for successful observing systems. Here and in following figures, the vertical lines measure the sensitivity to the choice of threshold for diagnosing high-mode emitters in the inversion.





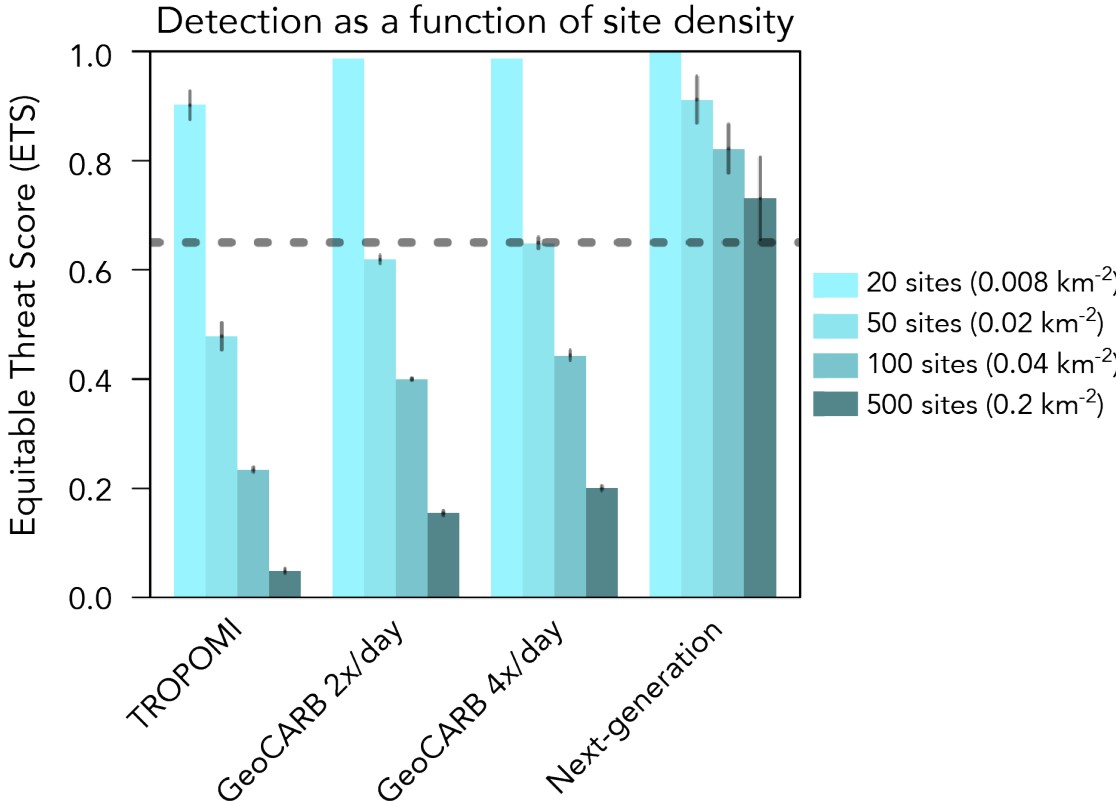

**Figure 6**. Equitable Threat Score (ETS) for each satellite observing configuration, varying the density of production sites
5    (20-500 sites in 50×50 km² domain). Results are from the L-1 inversion. The dashed line represents the ETS criterion for
successful observation.





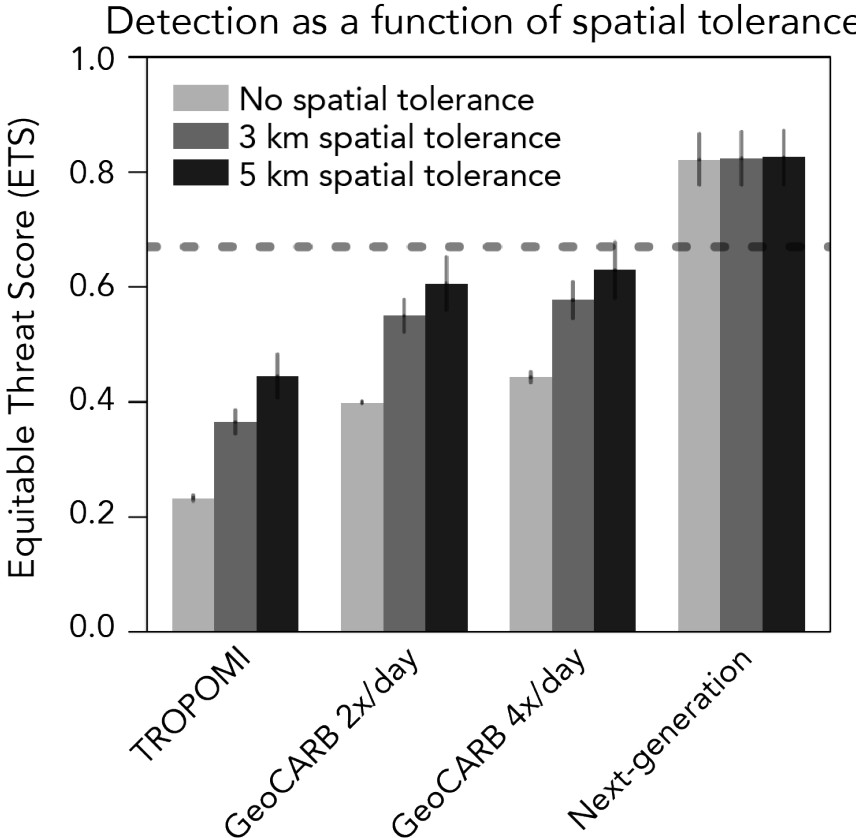

**Figure 7**: Effect of introducing spatial tolerance in the detection of high-mode emitters. Spatial tolerance is the radius within which a high-mode emitter must be located in order for a prediction to be called true positive (TP). The results are for an emission field with 100 production sites in the 50×50 km² domain. Only results from the L-1 inversion method are shown. The dashed line represents the ETS success criterion.





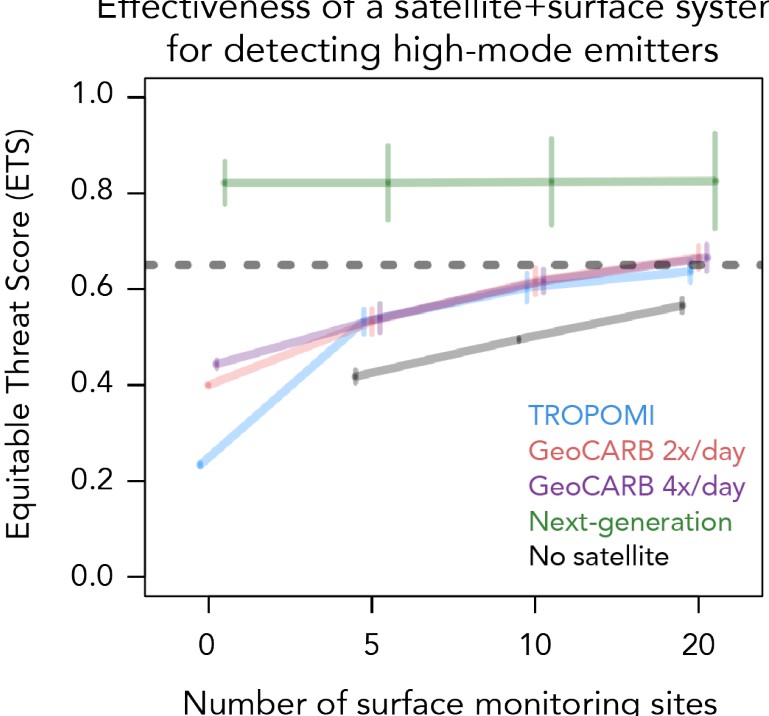

**Figure 8**: Effectiveness of a combined satellite and surface observing system for detecting high-mode emitters in an oil/gas field of 100 emitters over a 50×50 km$^2$ domain, as determined from joint inversion of the observations. The dashed line represents the ETS success criterion.