# Peer review of "Detecting high-emitting methane sources in oil/gas fields using satellite observations"

_Atmospheric Chemistry and Physics, 2018_

## Referee Comment (RC1) · Anonymous Referee #1 · 11 Sep 2018

**1   General comments**

The study addresses an interesting and current research topic, namely the monitoring of emissions from production sites in oil- and gas fields with different observing systems (satellite and in-situ). Even though these observation systems are not operated yet (with the exception of the TROPOMI instrument), it is important to investigate the capabilities of future measuring systems prior to their installation.

The study is clearly written and well structured; thus easy to follow. Also, the authors embed the study well in existing literature. Many references even for minor topics are provided.

[Figure]

The study uses modern concepts of inverse modeling, namely high resolution transport models and $L_1$-regularization. Although its success in other applications, $L_1$-regularization has rarely been used in environmental inverse modeling studies so far.

No data and code that can reproduce the given results are uploaded. Doing so would increase the value of the article and invite other research groups to contribute to the topic.

I recommend the publication after considering the following minor modifications.

**2   Specific comments**

Most of the article uses international standard units. However, the production rate of wells is described in $Mcf/d$ (in text, e.g. p. 3, ll. 12-13, and Fig. $2$ and $3$). I suggest to convert these to SI units.

The authors use '$a \times b$' to denote a scalar multiplication in some formulas (e.g., p. 6, l. 28; p. 7, l. 5; p. 8, l. 14), but not consistently. I recommend using '$a \cdot b$' or '$ab$' to be consistent with standard notation. To describe the dimensions of a matrix, $m \times n$ is the standard notation.

The phrase $L_1$-regularization originates from using the norm of the function space $L^p$ or the sequence space $l^p$ with $p = 1$. To be more consistent with the mathematical literature I recommend using the notation $L_1$ and $L_2$ instead of $L - 1$ and $L - 2$. It

would be beneficial to define the $L$-norm/$l^p$-norm in Eq. (1) (see p. 6, l. 1).

In Section $2.2$ (p. 4, ll. 20-26) I suspect there is a problem with the dimensions of $h_i$ and $x$. If $x \in \mathbb{R}^n$, then $h_i \in \mathbb{R}^n$ to build the scalar product. With this implicit definition $h_i$ changes for each realization of the scenario. On the other hand, $h_i$ (possibly $h_i$, $i = 1, 2, ...$) is defined as the *'archived footprint covering the complete set of observing locations and times'*. This definition seems to be incorrect. I think $h_i$ is the footprint corresponding to a particular measurement restricted to the locations of potential emitters. Then, the forward model $H$ for a particular configuration is build only by a subset of footprint indices that describe the corresponding measurements. There are several ways to define these quantities properly, but the way it is defined in the article seems incorrect.

As described in Sect. $1$ the Barnett Shale has $20000$ well pads in the $300\ km$ by $300\ km$ domain, i.e. a well density of $0.22\ wells/km^2$. Other oil fields, like the Kern River Oil Field near Bakersfield, CA, have much larger well densities ($> 200\ wells/km^2$). Are the chosen well densities representative for certain types of oil fields?
Also, if well pads (and other possibly emitting infrastructure) are not homogeneously distributed, the local density may be much larger. The densities analyzed in this study are thus more to the lower bound of what is required. The concept of spatial tolerance is an interesting extension. The analysis is carried out using the much lower well density ($0.04$). How do the results compare to the $0.2$-case? I expect that the results in Fig. 7 are too optimistic for many oil fields with densely distributed infrastructure.

Section $2.2$ describes how the pseudo-observations are created. It seems that no transport error is considered in the noise (p. 4, ll. 29-31). However, transport errors are mentioned when describing the inversion methods (p. 6, ll. 18-20). Are transport errors included in the study? I think they should!

The study considers column measurements by satellites but also a network of in-situ observations. The advantage of column measurements is that the in-flowing background concentration ($= b$, see p. 4., l. 26) is measured and the assumption that it is constant (or known) is justified. When considering only the in-situ network the background concentration is unknown, which is an additional challenge in the inversion. This aspect could be mentioned to support the assumption of a constant boundary in this study.

Also, I wonder at which altitude the in-situ analyzers are placed? I expect that local low mode emitters may have a significant influence on observations taken close to the surface.

In Sect. $2.5$ high-mode emitters are defined via the standard deviation, whereas in Sect. 2.1 high-mode emitters are those that exceed $40\ kg/h$. Which definition applies for the results? And what are the reasons to use that definition over the other?

The concept of $L_2$-regularization is well described in many textbooks covering inverse problems. I think that the given reference, i.e. Evgeniou et al. (2000), is not very helpful for applied atmospheric sciences. My recommendation for applied researchers would be *P. C. Hansen, Discrete Inverse Problems: Insight and Algorithms, 2010*, but many other options exist.

Using $L_1$-regularization to exploit the sparsity of the problem is a great idea, which turns out to give better results than the standard approach. This concept has rarely been used in atmospheric inverse modeling studies and is probably new to many in the research community. The references provided are helpful. Still, some questions

remain:

- How does a solution produced by $L_1$-regularization differ from one produced by $L_2$-regularization? The answer is described in Sect. 3.1, but I think a figure comparing both solutions for some representative realization would be useful. The same figure could also be used to illustrate how high-mode emitters are detected from the emission estimate.

- I assume and I hope that $5\%$ of high-mode emitters is generally a large estimate of failing systems. Further, I suspect that the identification of high-mode emitters improves for a smaller percentage of failing systems. Are there consequences on the solution produced by $L_1$-regularization if the solution is less sparse, i.e. more high-mode emitters? Is a low degree of sparsity important for the algorithm to perform well?

It could be argued that $20\%$ failure of an alarm system is still a lot, but some criteria for success needs to be applied. However, this criteria is neither mentioned in the abstract nor in the conclusions, when systems are defined successful or not. I think it should be briefly mentioned in both sections.

I recommend uploading scripts that reproduce the given results. The study could serve as a test environment for new modeling approaches and as an interesting project for students.

**3  Technical corrections**

P. 4, l. 2: A comma is missing in *'(small, medium, large)'*.

P. 4, l. 20: ... $1.3 \times 1.3 \ km^2$ pixels

P. 5, l. 28: remove brackets around $\hat{x}$

---

## Referee Comment (RC2) · Anonymous Referee #2 · 24 Sep 2018

The authors investigated the potential of using satellite observed XCH4 accompanied with surface observations to detect and locate high-mode emitting sites in oil/gas fields. They conducted OSSE on the basis of pseudo-observations from multiple satellites including both recently launched (TROPOMI) and the planned ones (GeoCARB and the next-generation geostationary one with a much finer resolution). An inverse approach was used to relate these satellite observations to the methane sources upwind, and to further assess the capability of different satellite data to detect high emission sites. The main results of this study suggest that the TROPOMI and the planned GeoCARB are unsuccessful at locating high-emitting sources in dense fields of >50 emitters within the $50 \times 50$ km2 domain. To address this issue, we need the next-generation geostationary satellite data that have a finer spatial-temporal resolution and a higher data precision,

or complement the current satellite data with a surface observation network. Overall, I think the paper reads well, provides interesting results, and fits the ACP scope. I have some concerns about the method part that should be addressed before this paper accepted for publication. My comments are as follows.

1. I have significant concerns about the effect of meteorological conditions (especially wind speed) on the detection of methane plumes by satellite observations. The high-wind condition promotes the dispersal and dilution of methane in the atmosphere, which makes the methane enhancement relative to background smaller than that under a low-wind condition, and the resulting low concentrations are much difficult to be detected by satellite. Therefore I wonder if the results of this paper are sensitive to the meteorological conditions used. This study was performed based on only a 1-week simulation using the WRF-STILT model, however, it used a very strong statement on the findings in the abstract and conclusion parts. It cannot convince me that the 7 days meteorological fields are representative enough. I suggest that the authors give a detailed discussion on the potential influence of meteorological conditions on their results.

2. It's not clear to me how this study used the WRF-STILT model because the model configurations are not described in details in Sect. 2.2. How many theoretical particles are released from each receptor, and what are the receptor heights? How is the column footprint calculated? Is it integrated from the footprints of different vertical layers? It is important that the method part is self-contained and does not require the reader to go through another source.

3. In the inverse method, the observational error covariance matrix accounts for instrument and model transport errors (line 18, page 6). What about the representation error? How is it considered for the satellite observations that have a different pixel size from the resolution of transport model?

---

## Referee Comment (RC3) · Anonymous Referee #3 · 26 Sep 2018

The authors have conducted observation system simulation experiments (OSSEs) to examine the potential of satellites to detect methane emissions from a dense distribution of industrial point sources. They compared the utility of data from TROPOMI, Geo-CARB, and a next-generation satellite instrument. They also compared the utility of the satellite data to that from a surface observing network. They found that TROPOMI and GeoCARB can detect high-mode CH4 emitters if the density of the emitters is low. Only the next-generation satellite would have the capability to detect a dense distribution of high-mode CH4 emitters. They suggested that combining TROPOMI or GeoCARB with surface data would help augment the detection capability of these instruments, whereas doing so would offer little additional benefit to the next-generation instrument. This is an interesting study and the manuscript is well written. I especially appreciate

the investigation of the impact of the L-1 and L-2 regularization. However, I do have a few major concerns. I would recommend publication of the manuscript in ACP after the authors have revised the manuscript to adequately address my concerns.

Major Concerns

1) My first major concern is regarding the treatment of clouds. The authors claim that they "assume clear-sky conditions to simplify the discussion" but this is a serious assumption, which, in my opinion, is unacceptable. The caveat at the end of the conclusions that states "as long as skies are clear" is really problematic. Clouds have a major impact on observational coverage. Would accounting for clouds enhance the disparity between the next-generation satellite and TROPOMI and GeoCARB? Actually, my main concern here is whether accounting for clouds would reduce the POD of the next generation satellite (as envisaged here) to less than 0.8, which would mean that even such an instrument would be unable to detect dense high-emitting sources. It is critical that the authors account for the impact of clouds in their analysis.

2) My other major concern is with the treatment of model transport error. The authors assumed an error of 4 ppb for both the surface and satellite observations. However, that is not a justifiable assumption. The model transport errors at the surface, in the vicinity of point sources, will be very different from that in the CH4 column. Assuming the same transport errors for these two types of measurements does not allow for a fair and meaningful comparison of the satellite and surface measurements.

3) My third concern is with the lack of discussion of the impact of systematic errors in the satellite data. Since the launch GOSAT it has become clear that systematic errors in the greenhouse gas retrievals pose a major challenge for the use of the data. I appreciate that it would be challenging for the authors to reasonably address the issue of biased retrievals in their OSSEs, but they should at least add some discussion in the manuscript about how systematic errors could confound the detection of the CH4 sources.

Technical Comments

1) Page 2, line 8: Satellites do not measure the atmospheric columns of methane. They measure backscattered solar radiation from which the atmospheric columns of methane are retrieved. Please change the wording here.

2) Table 1: The row is not properly aligned for the TROPOMI entry.

---

## Author Comment (AC1) · 16 Oct 2018

Response to Comments from Anonymous Referee #1

1. No data and code that can reproduce the given results are uploaded. Doing so would increase the value of the article and invite other research groups to contribute to the topic.

We thank the reviewer for this suggestion. A worked through script will be included with the publication. However, we will leave it to the reader to generate WRF-STILT footprints as the model is open source.

We point the reader to the location of the code on page 12, line 8:

[Figure]

Data Availability.     The WRF-STILT model is available for download at https://uataq.github.io/stilt/. A worked through example of the high-mode detection observing system simulation experiment (OSSE) described in this paper is available in the Supplementary Information for this paper.

2. Most of the article uses international standard units. However, the production rate of wells is described in Mcf/d (in text, e.g. p. 3, ll. 12-13, and Fig. 2 and 3). I suggest to convert these to SI units.

We add the unit conversion in the text:

Page 3, Line 11-12. (small: 10-100 million cubic feet per day (Mcf/d) where 1 Mcf/d = 0.028 Mm3/d; medium: 100-1000 Mcf/d; large: 1000+ Mcf/d)

3. The authors use 'a x b' to denote a scalar multiplication in some formulas (e.g., p. 6, l. 28; p. 7, l. 5; p. 8, l. 14), but not consistently. I recommend using 'a " b' or 'ab' to be consistent with standard notation. To describe the dimensions of a matrix, m x n is the standard notation.

We update the equations to be consistent with standard notation:

Page 7, Line 16. $r\_ij = \sigma\_M^2 \exp\{-d/l\}\exp\{-t/\tau\}$ for $i \neq j$ (5b)

Page 7, Line 25. $r\_ij = cor(i,j)\ \sigma\_Mi\ \sigma\_Mj\ \exp\{-d/l\}\exp\{-t/\tau\}$ (5c)

Page 9, Line 9. $\alpha = (\Sigma\ TP + \Sigma\ FP)(\Sigma\ TP + \Sigma\ FN)/(\Sigma\ TP + \Sigma\ FP + \Sigma\ FN + \Sigma\ TN) = 1/N\ (\Sigma\ FP)/FAR\ (\Sigma\ TP)/POD$

4. The phrase L1-regularization originates from using the norm of the function space Lp or the sequence space lp with p = 1. To be more consistent with the mathematical literature I recommend using the notation L1 and L2 instead of L-1 and L-2. It would be beneficial to define the L-norm/lp-norm in Eq. (1) (see p. 6, l. 1).

We include the standard definition of the L-norm in the text: Page 6, Line 20. The second term represents an adjustable parameter $\lambda$ and the L-norm of x, which is a

measure of the magnitude of the vector x defined as the following:

‖x‖_L= √(L&Σ_(k=1)ˆn |x_k |ˆL ) (2)

We change L-1 and L-2 to L1 and L2 everywhere else in the text.

5. In Section 2.2 (p. 4, ll. 20-26) I suspect there is a problem with the dimensions of hi and x. If x Rn, then hi Rn to build the scalar product. With this implicit definition hi changes for each realization of the scenario. On the other hand, hi (possibly hi, i = 1, 2,...) is defined as the 'archived footprint covering the complete set of observing locations and times'. This definition seems to be incorrect. I think hi is the footprint corresponding to a particular measurement restricted to the locations of potential emitters. Then, the forward model H for a particular configuration is build only by a subset of footprint indices that describe the corresponding measurements. There are several ways to define these quantities properly, but the way it is defined in the article seems incorrect.

We clarify the in the text:

Page 4, Line 28. We select the footprint information that corresponds to the locations of the n emitters so that hi is also a vector of n dimension.

6. As described in Sect. 1 the Barnett Shale has 20000 well pads in the 300 km by 300 km domain, i.e. a well density of 0.22 wells/km2. Other oil fields, like the Kern River Oil Field near Bakersfield, CA, have much larger well densities (> 200 wells/km2). Are the chosen well densities representative for certain types of oil fields? Also, if well pads (and other possibly emitting infrastructure) are not homogeneously distributed, the local density may be much larger. The densities analyzed in this study are thus more to the lower bound of what is required. The concept of spatial tolerance is an interesting extension. The analysis is carried out using the much lower well density (0.04). How do the results compare to the 0.2-case? I expect that the results in Fig. 7 are too optimistic for many oil fields with densely distributed infrastructure.

We thank the reviewer for these useful points on context and update the manuscript accordingly:

Page 10, Line 13. Actual fields can be even denser but we are limited in our investigation by the $1.3 \times 1.3$ km2 resolution of the WRF simulation.

7. Section 2.2 describes how the pseudo-observations are created. It seems that no transport error is considered in the noise (p. 4, ll. 29-31). However, transport errors are mentioned when describing the inversion methods (p. 6, ll. 18-20). Are transport errors included in the study? I think they should!

Non-perfect transport is anticipated and accounted for in the inversion through R (page 7, line 4). Transport errors are not included in the pseudo-observation creation (page 5, line 1), as this error is related to the instrument characteristics (Table 1).

8. The study considers column measurements by satellites but also a network of in-situ observations. The advantage of column measurements is that the in-flowing background concentration (= b, see p. 4., l. 26) is measured and the assumption that it is constant (or known) is justified. When considering only the in-situ network the background concentration is unknown, which is an additional challenge in the inversion. This aspect could be mentioned to support the assumption of a constant boundary in this study.

We address this issue in the text:

Page 5, Line 26. We assume that these sites report hourly data with 1 ppb precision and that the background concentration in surface air is constant, consistent with the assumption made for satellite observations. A variable background would complicate the problem but could be retrieved as part of the inversion (Wecht et al., 2014).

9. Also, I wonder at which altitude the in-situ analyzers are placed? I expect that local low mode emitters may have a significant influence on observations taken close to the surface.

We clarify in the text:

Page 4, Line 12. Surface observations are taken in the lowest model layer (centered at 28 m above ground) and the corresponding footprints are obtained by releasing and tracking back in time 100 particles at the observation location and time.

10. In Sect. 2.5 high-mode emitters are defined via the standard deviation, whereas in Sect. 2.1 high-mode emitters are those that exceed 40 kg/h. Which definition applies for the results? And what are the reasons to use that definition over the other?

We refer the reviewer to the discussion on page 8, line 10 of the manuscript:

"In a real-world application we would not know the actual pdfs of emissions (Figure 1), so we need to diagnose the occurrence of high-mode emitters on the basis of anomalies in the distribution of x ÌĆ."

We expand this discussion on page 8, line 13:

Using anomaly detection on x ÌĆ instead of a fixed threshold (e.g., 40 kg h-1) allows for generalization to other emission fields where the mean normal and high modes may be different than the Barnett Shale.

11. The concept of L2-regularization is well described in many textbooks covering inverse problems. I think that the given reference, i.e. Evgeniou et al. (2000), is not very helpful for applied atmospheric sciences. My recommendation for applied researchers would be P. C. Hansen, Discrete Inverse Problems: Insight and Algorithms, 2010, but many other options exist.

We thank the reviewer for the suggestion and change the reference to Hansen (2010).

12. Using L1-regularization to exploit the sparsity of the problem is a great idea, which turns out to give better results than the standard approach. This concept has rarely been used in atmospheric inverse modeling studies and is probably new to many in the research community. The references provided are helpful. Still, some questions

remain: - How does a solution produced by L1-regularization differ from one produced by L2-regularization? The answer is described in Sect. 3.1, but I think a figure comparing both solutions for some representative realization would be useful. The same figure could also be used to illustrate how high-mode emitters are detected from the emission estimate. - I assume and I hope that 5% of high-mode emitters is generally a large estimate of failing systems. Further, I suspect that the identification of high-mode emitters improves for a smaller percentage of failing systems. Are there consequences on the solution produced by L1-regularization if the solution is less sparse, i.e. more high-mode emitters? Is a low degree of sparsity important for the algorithm to perform well?

We include a new figure (now Figure 5) and add the following text:

Page 8, Line 4. Figure 5 shows the distribution x ÌĆ from a single realization of emissions, GeoCARB 4×/day pseudo-observations, and both L1 and L2 regularization. In this simulation, L1 regularization enables the retrieval of high-mode emitters while L2 regularization is more restrictive in allowing excursions from the low-mode mean.

Page 8, Line 15. Figure 5 shows thresholds for classifying high-mode emitters using anomaly detection and a fixed value of 40 kg h-1. The L1 threshold is larger than the L2 threshold, but smaller than the 40 kg h-1. Had the fixed threshold been used, some high-mode emitters (relative to x ÌĆ) would have not been classified as such.

13. It could be argued that 20% failure of an alarm system is still a lot, but some criteria for success needs to be applied. However, this criteria is neither mentioned in the abstract nor in the conclusions, when systems are defined successful or not. I think it should be briefly mentioned in both sections.

Page 1, Line 24. "...are successful (>80% detection rate, <20% false alarm rate) at locating high-emitting sources..."

Page 11, Line 20. GeoCarb shows little difference in success rate (Equitable Threat

Score (ETS) > 0.65) for 2 or 4 overpasses per day.

14. I recommend uploading scripts that reproduce the given results. The study could serve as a test environment for new modeling approaches and as an interesting project for students.

See Response to Comment #1.

15. P. 4, l. 2: A comma is missing in '(small, medium, large)'.

Fixed

16. P. 4, l. 20: ... 1.3 x 1.3 km2 pixels

Fixed

17. P. 5, l. 28: remove brackets around x Ì Ć

Fixed

Response to Comments from Anonymous Referee #2

1. I have significant concerns about the effect of meteorological conditions (especially wind speed) on the detection of methane plumes by satellite observations. The high-wind condition promotes the dispersal and dilution of methane in the atmosphere, which makes the methane enhancement relative to background smaller than that under a low-wind condition, and the resulting low concentrations are much difficult to be detected by satellite. Therefore I wonder if the results of this paper are sensitive to the meteorological conditions used. This study was performed based on only a 1-week simulation using the WRF-STILT model, however, it used a very strong statement on the findings in the abstract and conclusion parts. It cannot convince me that the 7 days meteorological fields are representative enough. I suggest that the authors give a detailed discussion on the potential influence of meteorological conditions on their results.

We thank the reviewer for bringing this point to light. We address this issue on page 5, line 6:

The mean daytime 10 m horizontal wind speed inside the observing domain during the simulated week is 5.4 m/s. Stronger winds could further dilute plumes within an observing domain, making the ability for satellite detection of emitters more difficult; on the other hand, the model transport error is less for stronger winds (Varon et al., 2018).

Page 11, Line 15. "Our results in these meteorological conditions can be summarized usefully in terms of answers to questions that a field manager might have:"

2. It's not clear to me how this study used the WRF-STILT model because the model configurations are not described in details in Sect. 2.2. How many theoretical particles are released from each receptor, and what are the receptor heights? How is the column footprint calculated? Is it integrated from the footprints of different vertical layers? It is important that the method part is self-contained and does not require the reader to go through another source.

We add more information in the text:

Page 4, Line 9. Footprints for each column were obtained by releasing and tracking back in time 100 particles from vertical levels centered at 28 m, 97 m, 190 m, 300 m above ground, and 8 additional levels up to 14 km altitude spaced evenly on a pressure grid.

3. In the inverse method, the observational error covariance matrix accounts for instrument and model transport errors (line 18, page 6). What about the representation error? How is it considered for the satellite observations that have a different pixel size from the resolution of transport model?

We clarify in the text:

Page 7, Line 5. Representation errors are negligible due to the model grid resolution finer or the same resolution as the instrument pixels (Turner et al., 2018).

[Figure]

Response to Comments from Anonymous Referee #3

1) My first major concern is regarding the treatment of clouds. The authors claim that they "assume clear-sky conditions to simplify the discussion" but this is a serious assumption, which, in my opinion, is unacceptable. The caveat at the end of the conclusions that states "as long as skies are clear" is really problematic. Clouds have a major impact on observational coverage. Would accounting for clouds enhance the disparity between the next-generation satellite and TROPOMI and GeoCARB? Actually, my main concern here is whether accounting for clouds would reduce the POD of the next generation satellite (as envisaged here) to less than 0.8, which would mean that even such an instrument would be unable to detect dense high-emitting sources. It is critical that the authors account for the impact of clouds in their analysis.

We thank for the reviewer for bringing up this important point. We expand upon our decision to consider only clear-sky on page 5, line 13.

Successful methane retrievals from satellites require clear sky. The probability of clear sky in a partly cloudy domain depends greatly on pixel size (Remer et al., 2012). Results for a partly cloudy condition would depend on the particular cloud configuration and would be difficult to generalize. Here we assume clear-sky conditions to avoid this complication, but the detection probability for high-mode emitters should then be viewed as an upper limit. In particular, it should be recognized that no detection from satellite is possible for a cloudy domain.

We explain how it relates to denser fields on page 10, line 13.

Actual fields can be even denser but we are limited in our investigation by the $1.3 \times 1.3$ km2 resolution of the WRF simulation.

2) My other major concern is with the treatment of model transport error. The authors assumed an error of 4 ppb for both the surface and satellite observations. However, that is not a justifiable assumption. The model transport errors at the surface, in the

vicinity of point sources, will be very different from that in the CH4 column. Assuming the same transport errors for these two types of measurements does not allow for a fair and meaningful comparison of the satellite and surface measurements.

We update our results and figures to account for this discrepancy:

Page 7, Line 10. "...$\sigma$M is the model transport error standard deviation previously estimated to be 4 ppb for methane columns (Turner et al., 2018). Given the order of magnitude difference in sensitivity between satellite columns and surface measurements (Figure 3), we assume $\sigma$M to be 40 ppb for surface measurements."

Page 7, Line 25. r_ij= cor(i,j) $\sigma$_Mi $\sigma$_Mj exp{-d/l}exp{-t/$\tau$} (5c)

We update Figures 6 and 9 (previously 5 and 8). The new analysis did not change conclusions in the text.

3) My third concern is with the lack of discussion of the impact of systematic errors in the satellite data. Since the launch GOSAT it has become clear that systematic errors in the greenhouse gas retrievals pose a major challenge for the use of the data. I appreciate that it would be challenging for the authors to reasonably address the issue of biased retrievals in their OSSEs, but they should at least add some discussion in the manuscript about how systematic errors could confound the detection of the CH4 sources.

We add more information about systematic errors in the text:

Page 5, Line 3. SWIR instruments may also suffer from systematic errors but we do not account for those here in the absence of information. The largest source of systematic error on our scale would likely be the inhomogeneity in surface reflectivity (Pfister et al., 2005).

Technical Comments 1) Page 2, line 8: Satellites do not measure the atmospheric columns of methane. They measure backscattered solar radiation from which the atmospheric columns of methane are retrieved. Please change the wording here.

We fixed the wording in the text:

Page 2, Line 8. Satellites measure backscattered solar radiation in the shortwave infrared (SWIR) from which atmospheric columns of methane can be retrieved with near uniform sensitivity down to the surface under clear-sky conditions (Jacob et al., 2016).

2) Table 1: The row is not properly aligned for the TROPOMI entry.

Fixed

Please also note the supplement to this comment:
https://www.atmos-chem-phys-discuss.net/acp-2018-741/acp-2018-741-AC1-supplement.zip
* * *

---

## Referee Report (RR1)

**Referee comment**

**anonymous author**

**1 General comments**

The authors addressed all concerns and questions raised by the reviewers. Most questions were answered by minor modifications of the manuscript. Only some aspects led to adaptations of the simulation setup. The modifications are well integrated in the previous version of the manuscript. The manuscript improved from these changes because the assumptions and limitations of the study are much better discussed.

A few comments remain and should be clarified prior to publication.

**2 Specific comments**

The supplement provides code written in R programming language, which is open source and easily applicable. The example code includes instructions, but is not running because the authors do not provide the required inputs, i.e. a forward operator or the footprints. The argument that the input can also be created using open source software is weak from my point of view, because installing and running a complex open source model, which has its own requirements, is far more an obstacle than running an example script with prepared inputs. I suggest to include the required input data as part of the supplement. I do not expect a full program, but a script that can reproduce some basic results.

From authors response:
*'7. Section 2.2 describes how the pseudo-observations are created. It seems that no transport error is considered in the noise (p. 4, ll. 29-31). However, transport errors are mentioned when describing the inversion methods (p. 6, ll. 18-20). Are transport errors included in the study? I think they should!*

*Non-perfect transport is anticipated and accounted for in the inversion through R (page 7, line 4). Transport errors are not included in the pseudo-observation creation (page 5, line 1), as this error is related to the instrument characteristics (Table 1).'*

From my point of view the transport errors should also be included in the creation of synthetic measurements (cp. e.g. Michalak et al., 2004: A geostatistical approach to surface flux estimation of atmospheric trace gases; Miller et al., 2014: Atmospheric inverse modeling with known physical bounds: an example from trace gas emissions). In a real data scenario the mismatch between simulated and measured data is not limited to instrumental noise. The noise model also includes (transport) model errors.

This aspect is important because the cross validation chooses a suitable weighting parameter $\lambda$ for the inversion method based on the input data. A small noise (instrumental error only) on the input data results in a smaller optimal weighting parameter and an improved reconstruction. This approach is used in the study and likely produces results that are too optimistic.

Estimation of the transport model error is a challenge by itsself but the assumption has already been made by the authors by the choice of the covariance matrix $R$. A higher level of noise (transport and instrumental error) forces the inversion with a larger weighting parameter and shows a reconstruction ability that is closer of what can be expected from (future) observation systems. The situation in such an approach is still optimistic because the noise has the same characteristics as anticipated by the covariance matrix $R$ in the reconstruction method.

If the intent of this study is to assume a perfect transport model because models may improve similar to observing systems the matrix $R$ should reflect the reduced transport error estimate. To me omitting the transport error reduces the value of the results significantly. Such an assumption should be stated clearly in the abstract and the conclusion.

Page 4, line 20: I would not use the phrase 'dilution effect' and rather explain in half a sentence why the surface influence is smaller for column measurements.

Page 4, lines 26ff: In the definition of the footprint $h_i$, i.e. $h_i = (\partial y_i/\partial x)^T$, the variable $x$ is already limited to the location of emitters and has dimension $n$. Thus, $h_i$ has dimension $n$. The authors continue: 'We select the footprint information that corresponds to the locations of the $n$ emitters so that $h_i$ is also a vector of $n$ dimension.'

To me this formulation seems a bit confusing. I suggest something like: '$h_i$ is then a vector of dimension $n$ with the selected footprint information that corresponds to the locations ...'.

Page 4, lines 30-32: The background $b$ is assumed to be constant in line 30. Implicitly, it is assumed to be zero in line 32. This discrepancy should be fixed.

Decide for one of the spellings: '$L_1$-regularization' or '$L_1$ regularization'. Both versions are used throughout the manuscript.

Page 6, line 13: I recommend using '(e.g. Hansen, 2010)' instead of '(Hansen, 2010)' , because it is just one of many possible sources.

Page 6, line 29: 'Evgeniou et al., 2000' does not appear in the literature list. Should it be replaced with 'Hansen, 2010' or something from the statistical literature?

Page 1, line 11 and page 11, lines 12-13.: As pointed out in Referee Comment #1 oil fields with much higher well densities exist. Since most of the analysis is carried out using the 100 emitter scenario, I am not convinced that such fields should be called dense (e.g. 'a high density of wells'). Are they dense for a particular type of gas fields (e.g. fracking)? Then, this should be specified.

Figure 5: An axis break at the y-axis could be useful. Including the thresholds is a great idea to vizualize the concept of detecting high-mode emissions. Which value of $S$ is used for the threshold? Maybe the uncertainties in the threshold by varying $S$ could be included in the figure, too.

Figure 9: Why is the uncertainty from the threshold increasing for a larger number of surface sites for the next-generation instrument?

Overall, the manuscript reads well and is an interesting contribution to the scientific community.

---

## Author Response (AR2)

We thank the reviewers for their thoughtful comments, which we have addressed below. All page and line numbers refer to those in the revised manuscript. Reviewer comments are in *italics*, our response is in plain text, and text in the revised manuscript is in blue.

**Response to Comments from Anonymous Referee #1**

1. *The supplement provides code written in R programming language, which is open source and easily applicable. The example code includes instructions, but is not running because the authors do not provide the required inputs, i.e. a forward operator or the footprints. The argument that the input can also be created using open source software is weak from my point of view, because installing and running a complex open source model, which has its own requirements, is far more an obstacle than running an example script with prepared inputs. I suggest to include the required input data as part of the supplement. I do not expect a full program, but a script that can reproduce some basic results.*

We include in the supplement an end-to-end worked example script with example data.

2. *From authors response: '7. Section 2.2 describes how the pseudo-observations are created. It seems that no transport error is considered in the noise (p. 4, ll. 29-31). However, transport errors are mentioned when describing the inversion methods (p. 6, ll. 18-20). Are transport errors included in the study? I think they should! Non-perfect transport is anticipated and accounted for in the inversion through R (page 7, line 4). Transport errors are not included in the pseudo-observation creation (page 5, line 1), as this error is related to the instrument characteristics (Table 1).' From my point of view the transport errors should also be included in the creation of synthetic measurements (cp. e.g. Michalak et al., 2004: A geostatistical approach to surface flux estimation of atmospheric trace gases; Miller et al., 2014: Atmospheric inverse modeling with known physical bounds: an example from trace gas emissions). In a real data scenario the mismatch between simulated and measured data is not limited to instrumental noise. The noise model also includes (transport) model errors. This aspect is important because the cross validation chooses a suitable weighting parameter for the inversion method based on the input data. A small noise (instrumental error only) on the input data results in a smaller optimal weighting parameter and an improved reconstruction. This approach is used in the study and likely produces results that are too optimistic. Estimation of the transport model error is a challenge by itsself but the assumption has already been made by the authors by the choice of the covariance matrix R. A higher level of noise (transport and instrumental error) forces the inversion with a larger weighting parameter and shows a reconstruction ability that is closer of what can be expected from (future) observation systems. The situation in such an approach is still optimistic because the noise has the same characteristics as anticipated by the covariance matrix R in the reconstruction method. If the intent of this study is to assume a perfect transport model because models may improve similar to observing systems the matrix R should respect the reduced transport error estimate. To me omitting the transport error reduces the value of the results signicantly. Such an assumption should be stated clearly in the abstract and the conclusion.*

We clarify that the same transport model was used for pseudo-observations and the inverse model.

Page 6, Line 10. "We use the same matrix **H** for both pseudo-observation construction and the inversion."

We clarify that transport error is added and accounted through **R**.

Page 7, Line 5. "The observational error covariance matrix $\mathbf{R} = (r_{ij})$ adds and accounts for both instrument and model transport errors."

3. *Page 4, line 20: I would not use the phrase 'dilution effect' and rather explain in half a sentence why the surface influence is smaller for column measurements.*

We change the text accordingly:

Page 4, Line 20. "Column footprints are about an order of magnitude smaller than surface footprints because surface signal is weakened for receptors (e.g., satellites) with total column sensitivity."

4. *Page 4, lines 26: In the definition of the footprint hi, i.e. hi = (dyi/dx)T , the variable x is already limited to the location of emitters and has dimension n. Thus, hi has dimension n. The authors continue: 'We select the footprint information that corresponds to the locations of the n emitters so that hi is also a vector of n dimension.' To me this formulation seems a bit confusing. I suggest something like: 'hi is then a vector of dimension n with the selected footprint information that corresponds to the locations ...'.*

We thank the reviewer for the suggestion and update the text.

Page 4, Line 28. "The vector $\mathbf{h}_i$ is also a vector of n dimension."

5. *Page 4, lines 30-32: The background b is assumed to be constant in line 30. Implicitly, it is assumed to be zero in line 32. This discrepancy should be fixed.*

We fix the discrepancy:

Page 4, Line 32. "$\mathbf{y}_{\mathbf{true}} = \mathbf{Hx} + b$"

6. *Decide for one of the spellings: 'L1-regularization' or 'L1 regularization'. Both versions are used throughout the manuscript.*

We decide on $L_1$ regularization and update accordingly in the manuscript.

7. *Page 6, line 13: I recommend using '(e.g. Hansen, 2010)' instead of '(Hansen, 2010)' , because it is just one of many possible sources.*

We add this caveat:

Page 6, Line 14. "(e.g., Hansen, 2010)"

8. *Page 6, line 29: 'Evgeniou et al., 2000' does not appear in the literature list. Should it be replaced with 'Hansen, 2010' or something from the statistical literature?*

We thank the reviewer for catching this error and include the correct citation to the text.

Page 13, Line 16. "Evgeniou, T., Pontil, M. and Poggio, T.: Regularization networks and support vector machines. Advances in Computational Mathematics, 13, 1, https://doi.org/10.1023/A:1018946025316, 2000."

9. *Page 1, line 11 and page 11, lines 12-13.: As pointed out in Referee Comment #1 oil fields with much higher well densities exist. Since most of the analysis is carried out using the 100 emitter scenario, I am not convinced that such fields should be called dense (e.g. 'a high density of wells'). Are they dense for a particular type of gas fields (e.g. fracking)? Then, this should be specified.*

We change the language describing the characterization of 100-emitter fields:

Page 2, Line 29. "Given a  population of production sites…"

Page 10, Line 19. "of high-mode emitters in  fields of 100+ wells."

Page 11, Line 13. "emitters among a  field of individual point sources."

We continue to characterize 500-emitter fields as dense as they match the well density of the Barnett Shale (Lyon et al., 2015).

10. *Figure 5: An axis break at the y-axis could be useful. Including the thresholds is a great idea to vizualize the concept of detecting high-mode emissions. Which value of S is used or the threshold? Maybe the uncertainties in the threshold by varying S could be included in the figure, too.*

We clarify in the caption the S threshold used.

Figure 5 (caption): "Dashed lines represent the thresholds to classify an emitter as high-mode, determined either from the distribution $\hat{\mathbf{x}}$ ($S = 2$) or from a fixed prior value (here 40 kg h$^{-1}$)."

11. Figure 9: *Why is the uncertainty from the threshold increasing for a larger number of surface sites for the next-generation instrument?*

We add a short discussion of this fact in the text:

[revised manuscript text omitted]